# Upgrading Argan Shell Wastes in Wood Plastic Composites with Biobased Polyethylene Matrix and Different Compatibilizers

**DOI:** 10.3390/polym13060922

**Published:** 2021-03-17

**Authors:** Maria Jorda-Reolid, Jaume Gomez-Caturla, Juan Ivorra-Martinez, Pablo Marcelo Stefani, Sandra Rojas-Lema, Luis Quiles-Carrillo

**Affiliations:** 1Departamento de Materiales y Tecnologías, Asociación de Investigación de la Industria del Juguete, Conexas y Afines (AIJU), Av. de la Industria, 23, 03440 Ibi, Spain; mariajorda@aiju.es (M.J.-R.); sanrole@epsa.upv.es (S.R.-L.); 2Instituto de Tecnología de Materiales (ITM), Universitat Politècnica de València (UPV), Plaza Ferrándiz y Carbonell 1, 03801 Alcoy, Spain; jaugoca@epsa.upv.es; 3Instituto de Investigaciones en Ciencia y Tecnología de Materiales (INTEMA), Consejo Nacional de Investigaciones Científicas y Técnicas (CONICET), Universidad Nacional de Mar del Plata (UNMdP), Av. Colón 10850, Mar del Plata 7600, Argentina; pmstefan@fi.mdp.edu.ar

**Keywords:** argan shell particles, wood plastic composite, polyethylene, mechanical properties, compatibilization

## Abstract

The present study reports on the development of wood plastic composites (WPC) based on micronized argan shell (MAS) as a filler and high-density polyethylene obtained from sugarcane (Bio-HDPE), following the principles proposed by the circular economy in which the aim is to achieve zero waste by the introduction of residues of argan as a filler. The blends were prepared by extrusion and injection molding processes. In order to improve compatibility between the argan particles and the green polyolefin, different compatibilizers and additional filler were used, namely polyethylene grafted maleic anhydride (PE-g-MA 3 wt.-%), maleinized linseed oil (MLO 7.5 phr), halloysite nanotubes (HNTs 7.5 phr), and a combination of MLO and HNTs (3.75 phr each). The mechanical, morphological, thermal, thermomechanical, colorimetric, and wettability properties of each blend were analyzed. The results show that MAS acts as a reinforcing filler, increasing the stiffness of the Bio-HDPE, and that HNTs further increases this reinforcing effect. MLO and PE-g-MA, altogether with HNTs, improve the compatibility between MAS and Bio-HDPE, particularly due to bonds formed between oxygen-based groups present in each compound. Thermal stability was also improved provided by the addition of MAS and HNTs. All in all, reddish-like brown wood plastic composites with improved stiffness, good thermal stability, enhanced compatibility, and good wettability properties were obtained.

## 1. Introduction

In the last decades, an increasing concern about environmental issues related to the great use of petrochemical-derived plastics has been rising, as well as the necessity to reduce the carbon footprint associated with the production processes of those polymers. This trend is driving the industry toward the use of polymeric materials derived from natural resources, with the objective of moving away from petroleum. Moreover, this fact has propitiated the use of biodegradable polymers, some of which are capable of decomposing in composting conditions. At the same time, they exhibit very similar properties to those of their petrochemical counterparts [1]. One of the main drawbacks of plastics derived from natural sources is that they are more expensive than those from petrochemical origin. This leads to a direct rejection by industry. For this reason, incorporation of fillers and creation of new more economic materials is gaining increasing relevance.

Among those materials, “Wood plastic composites” (WPC) are becoming increasingly popular. This technology implies the addition of natural organic fillers obtained from wastes into polymer matrices, with the objective of decreasing their processing cost. At the same time, those fillers enhance their environmental value and vary their mechanical, physical and chemical properties [2]. Initially, only wood flour or sawdust were used as bio-fillers [3,4,5,6], but over time fillers from agroforestry waste or food industry have started to be used, such as orange peel [7,8], almond shell [9], pomegranate peel [10], or argan seed shell [11]. Those fillers are introduced in the structure of polymers like poly (lactic acid) (PLA), high density polyethylene (HDPE) or polyester [12,13,14]. Nowadays, WPC can be employed as a substitute of wood due to its appearance for the fabrication of indoor and outdoor furniture, decks pergolas, and so on [15,16].

In this sense, high density bio-polyethylene (Bio-HDPE) is one of the most interesting polymer matrices, as it combines its natural origin with the ease of processability of polyolefins. This polymer is produced by conventional polymerization of ethylene obtained from catalytic dehydration of bio-ethanol [17], which is extracted from natural sources, such as sugarcane [18]. Bio-HDPE possesses the same physical properties than its fossil-derived counterpart; particularly, good mechanical resistance and high ductility [19]. Injection-molded Bio-HDPE pieces have great application in packaging [20]. However, one of the main problems that limits the use of bio-fillers as reinforcing agents is their hydrophilic nature, which causes a poor interfacial adhesion with a hydrophobic (non-polar) matrix, such as Bio-HDPE. This provokes a deterioration in the mechanical properties of the material and a bad compatibility between filler and matrix [21,22].

Several studies have been carried out to improve the compatibilization between those fillers and polymeric matrices. On the one hand, some of the methods that have been applied to improve this compatibility include the modification of the surface of the filler particles from the waste, such as mercerization, benzoylation, acetylation, silanation, or graft copolymerization [23,24]. On the other hand, compatibilizing agents have also been used in order to improve the adhesion and affinity between filler and matrix. Additionally, they have also been utilized to increase the dispersion of the reinforcing particles within the matrix and give additional properties to the blend [25,26,27]. In this context, polyethylene-grafted maleic anhydride (PE-g-MA) is one of the most efficient compatibilizers. It acts as a chemical connection between ligno-cellulosic particles and polymeric chains, owing to its double functionality. First, the polyethylene fraction of PE-g-MA interacts with polymeric chains due to its chemical affinity, while the anhydride groups can react with the hydroxyl groups from the ligno-cellulosic structure in the organic particles by esterification. Thus, leading to an increase in the matrix–filler interaction and a positive effect in the dispersion of the particles and the stiffness of the blend [28,29,30]. However, ductile properties tend not to get better, because of the fragility given by those fillers when they form stress-concentrator aggregates in the matrix.

In recent years, vegetable oils have generated a great deal of interest as compatibilizing additives. Modified vegetable oils have been used as plasticizers, stabilizers, crosslinkers, compatibilizers, and so on. The use of epoxidized vegetable oils (EVOS) should be remarked, such as epoxidized soybean oil (ESBO) [31,32], epoxidized linseed oil (ELO) [33,34], or epoxidized palm oil (EPO) [35]. All of them have been tested in polymeric blends and composites, achieving positive compatibilization/plasticization effects, owing to the reactivity of the oxirane group. Maleinization is an interesting alternative chemical modification for vegetable oils. Maleinized linseed oil (MLO) has been widely used in WPC, for example, in a PBS matrix with almond shell flour [36]. Furthermore, the excellent ductile properties that this oil is capable of giving to composites have also been reported [13]. In addition, inorganic particles can also be introduced as reinforcing agents in polymeric matrices, such as carbon nanotubes (CNTs), nanoclays, or metallic oxide nanoparticles [37,38,39]. In recent years, halloysite nanotubes (HNTs) have gained great importance in this sense. They are cheaper and more abundant than, for example, carbon nanotubes (CNTs). In this sense, halloysite is a natural mineral clay with nanotube morphology, similar to kaolinite (Al_2_[Si_2_O_5_(OH)_4_].nH_2_O), where n is 2 for hydrated HNTs, and 0 for dehydrated HNTs [40]. Frost et al. [41] reported two types of hydroxyl groups, inner and outer hydroxyl groups, placed between the nanotubes layers and in the surface of HNTs [41]. Because of their multilayer structure, most of the hydroxyl groups are in the inner side, while a few of them are in the surface of HNTs. As a result of this, HNTs dispersion in polymer matrices is simpler than other inorganic nanoparticles [40]. Numerous studies have reported the ability of HNTs to improve thermal stability, fire resistance, mechanical properties, and crystalline behavior of polymers such as polypropylene [42,43,44,45,46], low density polyethylene (LLDPE) [47], polyamide 6 (PA6) [48], epoxy resin [49], polylactide (PLA) [50,51], and so on. These are the reasons for utilizing HNTs in this work. As it is the case with organic particles, the challenge here is to achieve a good compatibility and adhesion between HNTs and the polymer matrix, especially in non-polar matrices, such as polyolefins.

The objective of this work is the reuse of micronized argan shell (MAS) as a reinforcing organic filler in a Bio-HDPE matrix derived from sugarcane, with the objective of developing a wood plastic composite with enhanced properties compared to the original matrix following the trends proposed by the circular economy, which seeks to reuse or recycling the waste and to reintroduce them into the industry [52]. Argan (*Argania spinosa*) is a tropical plant that belongs to the *Sapotaceae* family, whose fruit is used in morocco to prepare oil [53]. This oil has great application in cosmetics owing to its high E vitamin content [53], while fruit residues and seeds are generally used to feed cattle [54]. The main drawback of micronized argan shell (MAS) is its low compatibility with Bio-HDPE. In particular, the challenge of this investigation is to improve the affinity between both elements through the use of compatibilizing agents. Several formulations have been developed to meet this end, using the compatibilizers PE-g-MA and MLO; and halloysite nanotubes (HNTs) as additional reinforcing fillers; altogether with Bio-HDPE as the polymer matrix and argan shell particles as the main reinforcing filler; these elements have been used both individually and in combination. Additionally, MLO and HNTs concentrations have been varied (7.5 parts per hundred resin (phr) when used individually and 3.75 phr when used in combination). In order to evaluate the effects of these compatibilizers and fillers over each blend, mechanical, morphological, thermal, thermomechanical, colorimetric, and wettability properties are presented.

## 2. Materials and Methods

### 2.1. Materials

Bio-HDPE, SHA7260 grade, was supplied in pellets form by FKuR Kunststoff GmbH (Willich, Germany) and manufactured by Braskem (São Paulo, Brazil). This green polyethylene has a density of 0.955 g cm^−3^ and a melt flow index (MFI) of 20 g/10 min, measured with a load of 2.16 kg and a temperature of 190 °C. Micronized argan shell (MAS) was supplied by MICRONIZADOS VEGETALES S.L. (Benamejí (Córdoba), Spain) company. Halloysite nanotubes were supplied by Sigma Aldrich (Madrid, Spain) with CAS number 1332-58-7.

Interlocking is one of the most important mechanisms in polymer composites. This mechanism depends on the filler shape. Figure 1a shows the morphology of the argan particles observed by SEM. Most of the particles exhibit a rough surface, which can be ascribed to the milling process due to the high hardness of this filler. A closer observation of the particles shows some porosity on their surface, which can produce a good interaction with the polymer matrix acting as mechanical anchoring points. A similar structure was observed by Laaziz et al. [55] when studying argan nut shell particles with different treatments. Additionally, Figure 1b and c show histograms for the length and diameter of the particles, respectively, determined from the SEM image. Particles were average 70 µm long and 45 µm wide. These parameters are also important to consider when talking about mechanical properties. Too large particles could lead to a great mechanical impairment, increasing the heterogeneity of the blends. Crespo et al. [56] observed this effect for almond shell particles superior to 150 µm diameter.

In relation to the additives, polyethylene-grafted maleic anhydride (PE-g-MA) with CAS Number 9006-26-2 and MFI values of 5 g/10 min (190 °C/2.16 kg), was obtained from Sigma-Aldrich S.A. (Madrid, Spain). This PE-based copolymer was selected due to its functionality. The maleinized linseed oil—MLO, VEOMER LIN was supplied by Vandeputte (Mouscron, Belgium). This modified vegetable oil is characterized by a viscosity of 10 dPa s at 20 °C and an acid value comprised in the 105–130 mg KOH g^−1^ range.

### 2.2. Preparation of Bio-HDPE Blends

Bio-HDPE, PE-g-MA, MAS y HNTs were initially dried at 40 °C for 48 h in a dehumidifying dryer MDEO (Barcelona, Spain) to remove any residual moisture prior to processing to avoid the possibility of hydrolysis due to the moisture. Then, the corresponding wt.% of each component (see Table 1) were mixed and pre-homogenized in a zipper bag. The corresponding formulations were compounded in a twin-screw co-rotating extruder from Construcciones Mecánicas Dupra, S.L. (Alicante, Spain). This extruder has a 25 mm diameter with a length-to-diameter ratio (L/D) of 24 to allow a correct blending process. The extrusion process was carried out at a rate of 22 rpm, using the following temperature profile (from the hopper to the die): 140–145–150–155 °C. The compounded materials were pelletized using an air-knife unit. In all cases, residence time was approximately 1 min. Table 1 shows the compositions of the formulations developed in this work.

To transform the pellets into standard samples, a Meteor 270/75 injector from Mateu & Solé (Barcelona, Spain) was used. The temperature profile in the injection molding unit was 135 °C (hopper), 140 °C, 150 °C, and 160 °C (injection nozzle). A clamping force of 75 tons was applied while the cavity filling and cooling times were set to 1 and 10 s, respectively. Standard samples for mechanical and thermal characterization with an average thickness of 4 mm were obtained.

### 2.3. Characterization of Bio-HDPE Blends

#### 2.3.1. Mechanical Characterization

Mechanical properties were obtained with different tests like tensile test, shore hardness, and Charpy impact test. Tensile properties of Bio-HDPE/PE-g-MA/MAS blends were obtained in a universal testing machine ELIB 50 from S.A.E. Ibertest (Madrid, Spain) as recommended by ISO 527-1:2012 with dog bone samples 1B specification. A 5-kN load cell was used and the cross-head speed was set to 5 mm/min. Shore hardness was measured in a 676-D durometer from J. Bot Instruments (Barcelona, Spain), using the D-scale, on rectangular samples with dimensions 80 × 10 × 4 mm^3^, according to ISO 868:2003. The impact strength was also studied on injection-molded rectangular samples with dimensions of 80 × 10 × 4 mm^3^ in a Charpy pendulum (1-J) from Metrotec S.A. (San Sebastián, Spain) on notched samples (0.25 mm radius V-notch), following the specifications of ISO 179-1:2010. All mechanical tests were performed at room temperature, and at least six samples of each material were tested and the corresponding values were averaged.

#### 2.3.2. Morphology Characterization

The morphology of fractured samples from Charpy tests, obtained from the impact tests, were studied by field emission scanning electron microscopy (FESEM) in a ZEISS ULTRA 55 microscope from Oxford Instruments (Abingdon, UK). Before placing the samples in the vacuum chamber, they were sputtered with a gold-palladium alloy in an EMITECH sputter coating SC7620 model from Quorum Technologies, Ltd. (East Sussex, UK). The FESEM was operated at an acceleration voltage of 2 kV.

#### 2.3.3. Thermal Analysis

The most relevant thermal transitions of Bio-HDPE/PE-g-MA/MAS blends were obtained by differential scanning calorimetry (DSC) in a Mettler-Toledo 821 calorimeter (Schwerzenbach, Switzerland). Samples with an average weight of 6–7 mg, were subjected to a thermal program divided into three stages: a first heating from 25 °C to 160 °C followed by a cooling to 0 °C, and a second heating to 250 °C. Both heating and cooling rates were set to 10 °C/min. All tests were run in nitrogen atmosphere with a flow rate of 66 mL/min using standard sealed aluminum crucibles with a capacity of 40 μL.

The thermal degradation of the Bio-HDPE/PE-g-MA/MAS blends was assessed by thermogravimetric analysis (TGA). TGA tests were performed in a LINSEIS TGA 1000 (Selb, Germany). Samples with a weight of 15–17 mg were placed in 70 µL alumina crucibles and subjected to a dynamic heating program from 40 °C to 700 °C at a heating rate of 10 °C/min in air atmosphere. The first derivative thermogravimetric (DTG) curves were also determined. All tests were carried out at least three times in order to obtain reliable results.

#### 2.3.4. Thermomechanical Properties

In order to obtain the thermomechanical properties of the Bio-HDPE composites a dynamical mechanical thermal analysis (DMTA) was carried out in a DMA1 dynamic analyzer from Mettler-Toledo (Schwerzenbach, Switzerland), working in single cantilever flexural conditions. Rectangular samples with dimensions 20 × 6 × 2.7 mm^3^ were subjected to a dynamic temperature sweep from −150 °C to 120 °C at a constant heating rate of 2 °C/min. The selected frequency was 1 Hz and the maximum flexural deformation or cantilever deflection was set to 10 µm.

#### 2.3.5. Color and Wetting Characterization

The colorimetric modifications of the samples were measured with a Konica CM-3600d Colorflex-DIFF2, from Hunter Associates Laboratory, Inc. (Reston, VA, USA) was used. Color coordinates (L*a*b*) were measured according to the following criteria: L* = 0, darkness; L* = 100, lightness; a* represents the green (a* < 0) to red (a* > 0); b* stands for the blue (b* < 0) to yellow (b* > 0) coordinate.

Hydrophilicity/ hydrophobicity of each blend was assessed by contact angle measurements through time were carried out with an EasyDrop Standard goniometer model FM140 (KRÜSS GmbH, Hamburg, Deutschland) which is equipped with a video capture kit and analysis software (Drop Shape Analysis SW21; DSA1). Double distilled water was used as test liquid, a drop of it was put in each sample and contact angle measurements were taken at 0, 5, 10, 15, 20, and 30 min after the administration of the water. For color and wetting measurement, traction samples were used to facilitate the measures.

#### 2.3.6. Water Absorption Test

The water absorption capacity of the Bio-HDPE/PE-g-MA/MAS blends was evaluated by the water uptake method. Impact specimens (80 × 10 × 4 mm^3^) from each blend were first weighted in a balance and then put inside a beaker filled with distilled water, all of them wrapped with tiny pieces of a metal grid so they could sink. After that, the weight of all samples was measured in intervals of several hours the first day, and then measured each week for 14 weeks in order to evaluate the amount of taken up water. In every measurement, the moisture in the surface of the samples was removed with tissue paper. During the immersion time, the temperature was controlled to 23 °C.

#### 2.3.7. Infrared Spectroscopy

In order to obtain the interactions between the different elements, a chemical analysis of the Bio-HDPE/PE-g-MA/MAS blends was carried out by attenuated total reflection-Fourier transform infrared (ATR-FTIR) spectroscopy. Spectra were recorded using a Bruker S.A Vector 22 (Madrid, Spain) coupled to a PIKE MIRacleTM single reflection diamond ATR accessory (Madison, WI, USA). Data were collected as the average of 10 scans between 4000 and 500 cm^−1^ with a spectral resolution of 2 cm^−1^. Samples from impact specimens (80 × 10 × 4 mm^3^) were employed in this test and the measurement was performed at room temperature.

#### 2.3.8. Statistical Analysis

The significant differences among the samples were evaluated at 95% confidence level (*p* ≤ 0.05) by one-way analysis of variance (ANOVA) following Tukey’s test. Software employed for this propose was the open source R software V4.0.3 (http://www.r-project.org (accessed on 12 March 2021))

## 3. Results

### 3.1. Mechanical Properties

The results concerning the mechanical properties of Bio-HDPE/MAS blends with different compatibilizers are shown in Table 2. Those results are of great interest to evaluate the effectivity of the different agents used in terms of resistance improvement of Bio-HDPE and compatibility between Bio-HDPE and MAS.

As it can be observed in Figure 2, neat Bio-HDPE presents a Young modulus (E) and tensile strength of 750 and 14.48 MPa, respectively. Elongation at break (ε_b_) could not be determined because the tensile test machine reached its maximum elongation without breaking the sample. Those values are indicative of certain stiffness and high ductility for Bio-HDPE, as it has been also reported by other authors [57]. Incorporation of MAS (30 wt.%) with PE-g-MA (3 wt.%) into the Bio-HDPE matrix increases the Young modulus up to 846 MPa, in this case the statical analysis showed a significant difference by the incorporation of the filler. This can be directly related to a proper distribution of MAS particles in the polymer matrix, leading to a good adhesion of the filler with the polymer and obtaining a more rigid material [26,27]. However, there is a clear reduction in tensile strength and elongation at break (Figure 2), this can be related to an excess of MAS utilized, increasing the stiffness of the material. In this context Essabir et al. [11] observed a similar behavior when treating HDPE with argan particles, in a way that Young modulus increased with MAS concentration, but tensile strength started to decrease with a MAS particle content superior to 5%. Addition of HNTs in the blend showed an additional enhancement of stiffness; the incorporation of only 7.5 phr HNTs produced a significant difference against the previous sample and a 33% of improvement. In this context, Bio-HDPE/PE-g-MA/MAS/HNTs sample presented the highest Young modulus of all blends (1126 MPa). This can be attributed to the reinforcing effect of HNTs, which seem to be well-dispersed within the matrix owing to the presence of the compatibilizer PE-g-MA. Similar behavior was reported by Pratap et. al. [58]. On the other hand, HNTs do not produce changes in tensile strength (no significant differences could be appreciated between them) and reduce elongation at break. Therefore, the material becomes less ductile, as a result of the combination of MAS and HNTs reinforcing effects. MLO-compatibilized blends saw their Young modulus and tensile strength reduced in comparison with the rest of the blends. Nevertheless, elongation at break increased 100% referring to the Bio-HDPE/PE-g-MA/MAS blend and the Tukey test showed a significant difference between the samples. In this sense, the plasticizing effect of MLO can be verified, highly increasing the ductile properties and compatibility between components in the blend. Similar results were observed by Quiles-Carrillo et. al. [59] when treating a PA1010/Bio-HDPE blend with MLO. As to Bio-HDPE/PE-g-MA/MAS/HNTs/MLO blend, it shows intermediate values between the sample with HNTs and the sample with MLO individually. Although the plasticizing effect of MLO seems to prevail over the reinforcing effect of HNTs, due to the very high elongation at break and the relatively low Young modulus and tensile strength.

Regarding the impact strength results, neat Bio-HDPE presents a value of 2.7 kJ/m^2^, which is a ductile behavior indicator. Bio-HDPE/PE-g-MA/MAS blend shows a reduction of impact strength down to 1.4 kJ/m^2^, confirming the great stiffness of the blend, probably due to the quantity of MAS utilized [11]. The presence of HNTs in the polymer matrix increases the impact strength to 0.3 kJ/m^2^, which means a significant difference between the mentioned samples. This effect is probably associated with an interaction between argan particles and HNTs. Moreover, it is possible that HNTs presence leads to a good dispersion of the argan particles, as well as HNTs themselves, boosting the reinforcing effect of the fillers [58]. As it was expected, MLO-compatibilized system shows a great impact strength due to the plasticizing effect of MLO. Several studies have reported the effectivity of MLO as a compatibilizing agent to increase the impact strength of fragile materials, such as PLA or PLA/almond shell flour blends [33]. Impact strength of HNTs/MLO combinated system is very similar to that of MLO blend, remarking the plasticizing effect of this oil. In this sense, the addition of HNTs to the composites with MLO did not provide a significant difference.

With regard to Shore D hardness, a similar trend to that observed in Young modulus (E) can be observed. HNTs sample shows the highest hardness (60.6), giving evidence of the reinforcing effect of nanotubes and MAS. A very similar value is displayed by the ternary blend Bio-HDPE/PE-g-MA/MAS (59.2), reflecting the hardening effect of MAS, composites without MLO did not show significant differences between them. As expected, MLO samples exhibit the lowest hardness (53.2), due to the great ductility this component bestows upon the blend. Ferri et. al. reported this same effect in PLA/TPS blends with different proportions of MLO [60]. The combination of HNTs and MLO leads to a slightly superior hardness to that of the MLO sample, this provoked by the hardening effect of HNTs and the reduction in MLO concentration (going from 7.5 phr to 3.75 phr). The ANOVA analysis showed that MLO had a clear effect on the hardness.

### 3.2. Morphology of Bio-HDPE/MAS Blends

Internal morphology of these materials is directly related to their mechanical properties. Figure 3 shows the field emission scanning electron microscopy (FESEM) images at 1000× of argan particles and the surface of fractured impact samples of each one of the blends. First, Figure 3a shows the morphology of an argan particle, which presents a rough surface, emphasized by the presence of holes on its structure (arrow). Those holes play an important role in the adhesion of the particles into the matrix according to Crespo et al. [56,61], who reported a very similar morphology in almond shell flour particles. Figure 3b corresponds to neat Bio-HDPE, which exhibits the typical irregular, rough, and cavernous surface of a ductile polymer, as it was also reported by Quiles-Carrillo et. al. [57]. Regarding the incorporation of argan particles, they show great cohesion within the matrix, as it is shown in Figure 3c–f. The gap between the perimeter of the particles and the polymer matrix is really narrow (arrows), which implies good interaction between both elements. This behavior can be well-related to the action of PE-g-MA as a compatibilizer and justifies the increase in stiffness, in terms of elastic modulus, of the samples that do not have MLO in their structure. Samples in Figure 3e,f show wire-shaped long structures in the polymer matrix (circles). They indicate a more ductile fracture and they confirm the plasticizing effect of MLO. This is directly related to the high elongation at break displayed in mechanical properties. Moreover, argan particles seem to be more immersed in the matrix in samples with MLO, revealing a contribution of the oil to compatibilization. This was also observed by Quiles-Carrillo et. al. [13], when adding MLO to PLA and almond shell flour blends. Additionally, with the presence of HNTs and MLO in the structure, a positive decrease in the number of gaps inside the matrix can be observed. Those gaps are originated by the fall of argan particles after fracture, thus being indicative of a worse interaction between MAS and Bio-HDPE.

Presence of HNTs in the blends can be observed in argan particles in Figure 3d,f. Figure 4a,b better illustrates them, where 5000× FESEM images of Bio-HDPE/PE-g-MA/MAS/HNTs and Bio-HDPE/PE-g-MA/MAS/HNTs/MLO samples are shown, respectively. In this context, a correct distribution of nanotubes can be observed in the surface of argan particles, which are indicated in the images by circles. This behavior could be attributed to a synergy between hydroxyl groups of HNTs and hydroxyl and anhydride groups present in PE-g-MA and lignocellulosic compounds of MAS [62]. As expected, nanotubes density is superior in the 7.5 phr HNTs sample (Figure 4a), where nanotubes cover almost all the argan particle surface. While in the 3.75 phr HNTs sample (Figure 4b), nanotubes are disposed in several agglomerates in the surface of MAS. This fact could be responsible for the low increase in resistance in 3.75 phr HNTs sample, altogether with the plasticizing effect of MLO.

### 3.3. Thermal Properties of Bio-HDPE/MAS Blends

Figure 5 gathers the DSC thermograms corresponding to the second heating cycle of the studied samples; Table 3 gathers melting temperature (T_m_) and crystallinity (X_c_) values of every Bio-HDPE blend. Regarding thermograms, in this case only melting temperature can be observed, because glass transition temperature is really low for Bio-HDPE (−100 °C) [63]. In the case of neat Bio-HDPE, it shows a melting temperature of 131 °C and a crystallinity X_C_ of 66.3%. Similar values were reported by Quiles-Carrillo et al. [64]. After the incorporation of argan particles alongside PE-g-MA into the Bio-HDPE, melting temperature undergoes a slight decrease down to 130.4 °C. On the other hand, crystallinity highly decreases to 49.7%. This could be due to the fact that, although argan particles can induce heterogeneous nucleation over the crystallization process of the polymer, high MAS concentrations (over 10% MAS) can compromise nucleation because of the particle–particle contact. Therefore, a limitation in space for crystal formation and growth is produced. This phenomenon was observed by Essabir et al. [65]. HNTs increase T_m_ up to 133.1 °C. This effect was also observed by Erding et al. [66] after studying the melting temperature of polyethylene and halloysite nanotubes composites (PE/HNTs). This can be ascribed to the fact that HNTs have some insulating properties against heat transfer, which is common in clay-based materials. Crystallinity suffers a great decrease in this case (37.6%), probably due to an excess of nucleating agents in the blend. This excess is originated by the combined presence of HNTs and argan particles. In this context, several studies have demonstrated the ability of HNTs to favor crystallization of polymer matrices, as long as there is no overload (+20 phr), to avoid the formation of aggregates that hinder the crystal nucleation and growth process [67]. Regarding MLO, it does not vary T_m_, but reduces, along with argan, crystallinity down to 39.8%. MLO improves compatibilization and, therefore, argan particle dispersion in the polymer matrix, thus contributing to reducing polymer–polymer interactions and favoring crystal formation [68]. However, in this case there are excess of argan particles, as it was aforementioned, that diminishes the crystallinity of the matrix. HNTs, MAS, and MLO sample hardly presents any change referring to the melting temperature and it reduces crystallinity in a very similar manner to that of MLO sample, due to obstruction of heterogeneous nucleation, as it has been previously explained. Regarding the statistical analysis of the variance, the melting temperatures did not showed modifications. A different trend could be observed when talking about the melting enthalpy. The first significant difference could be observed when the filler was introduced and as a result a dilution effect was produced on the blends. Depending on the compatibilization strategy of each material, the degree of crystallinity showed minor modifications as commented above.

Thermal stability has also been studied. Figure 6 shows both TGA curves and their first derivative (DTG), while Table 4 gathers the main quantitative parameters related to thermal degradation. Neat Bio-HDPE presented a temperature of approximately 345 °C for a mass loss of 5 wt.-% (T_5%_), while its maximum degradation rate temperature was (*T_deg_*) 447 °C. Additionally, after a single-phase degradation, its residual mass at 700 °C was 0.3%. Montanes et. al. [69] observed a similar thermal profile for neat Bio-HDPE. Addition of argan fillers and compatibilizers seem to have decreased the initial degradation temperature to values between 275 and 290 °C, the Tukey analysis showed a significant difference in these cases. This is mainly attributed to the presence of MAS in the polymer structure, which reduces in certain manner its thermal stability. This was also reported by Essabir et. al. [26] when they studied the properties of polypropylene and almond shell flour blends. Interestingly, it can be seen as all other samples undergo several degradation stages. Particularly, Bio-HDPE/PE-g-MA/MAS sample shows a three-phase degradation, attributed to the hemicellulose and pectin degradations (280–340 °C); cellulose degradation (340–448 °C) and lignin degradation (448–477 °C) which are present in argan particles [26,70]. These different phases appear in all samples except for neat Bio-HDPE. MLO samples seem to provide some more thermal stability to the blend, due to its capacity to link polymer chains (crosslinking), which is a positive effect toward stability. With regard to the maximum degradation rate temperature, it has increased in all cases except the MLO sample. This is highly related to the compatibilizing effect of PE-g-MA altogether with argan particles and HNTs [58], leading to a good particle dispersion and retarding the maximum degradation peak to 481.3 and 458.3 °C for Bio-HDPE/PE-g-MA/MAS and Bio-HDPE/PE-g-MA/MAS/HNTs samples, respectively. Referring to HNTs, their tubular hollow structure, which is capable of holding degradation products inside, is responsible for slowing the mass transport mechanism, as it was reported by Pratap et al. [58].

According to residual mass results, all samples have more residual mass than neat Bio-HDPE. This is especially logic with HNTs, as they are mostly inorganic and possess a very high degradation temperature [58]. In this case 7.5 phr HNTs sample retains a residual mass of 7.8 wt.-% while the sample with 3.75 phr HNTs maintains a 4.3 wt.-% residual mass. The introduction of the lignocellulosic fillers also provided an increasement of the residual mass due to the introduction of lignocellulosic compounds that cannot be degraded at 700 °C as Liminana et al. reported for the Almon Shell Flour [36]. Regarding the residual mass, all the samples show significant differences between the previous blend, this is manly by the incorporation of the HNT and the MAS that provide different residual mass as mentioned before.

### 3.4. Dynamic-Mechanical Behavior of Bio-HDPE/PE-g-MA/MAS Blends

Thermomechanical properties of blends were studied by DMTA technique. Figure 7a shows the evolution of the storage modulus (G’) from −150 to 120 °C. In relation to neat Bio-HDPE, an initial decrease in G’ can be observed until −100 °C approximately. This is directly connected with the glass transition of the material. Then, the storage modulus further decreased due to softening of the polymer matrix. Quiles-Carrillo et. al. [57] observed a very similar profile for this polyolefin. Addition of MAS and PE-g-MA into the matrix slightly increases G’ modulus along all the temperature range. Specifically, a rise is produced at −140 °C (see Table 5) from 2513 MPa (Bio-HDPE) to 2523 MPa (Bio-HDPE/PE-g-MA/MAS). This effect emphasizes as temperature decreases, going from 1309 to 1413 MPa at −25 °C. This is caused by a stiffening of the polymer due to the reinforcing effect of argan particles [65]. As expected, HNTs presence highly augments the storage modulus, providing values of 3111 and 1898 MPa at −140 and −25 °C (see Table 5), respectively, in comparison to the values obtained previously for neat Bio-HDPE. This is indicative of an increase in the stiffness of the material, confirming the results commented in the mechanical properties section, associated with a high dispersion of HNTs in the matrix. Thus provoking a decrease in polymer chain mobility due to physical interactions between nanotubes and adjacent chains [71]. In the case of Bio-HDPE/PE-g-MA/MAS/MLO sample, superior values of storage modulus were observed until 0 °C in comparison to Bio-HDPE and Bio-HDPE/PE-g-MA/MAS samples, but the modulus was inferior in the last section, till 120 °C. This reduction can be associated with the plasticizing effect of MLO. Although the difference is not significant and, given the modulus increase in the initial section of the diagram, it can be asserted that MLO acts as a dispersing element of the argan particles, in combination with PE-g-MA. Quiles-Carrillo et. al. [57] observed a similar behavior by studying the effect of MLO to compatibilize Bio-HDPE/PLA blends. Finally, the combined effect of HNTs (3.75 phr) with MLO (3.75 phr) enhances the stiffness of Bio-HDPE in all the temperature range, but to a lesser extent than that of 7.5 phr HNTs sample, as the concentration is lower in this case. Nonetheless, a good compatibilization and dispersion effect of MLO toward HNTs and MAS can be seen. The statistical analysis shows a similar trend of the storage modulus with respect to the elastic modulus.

Figure 7b shows the evolution of the damping coefficient (tan δ) with temperature for every studied blend. The peak observed at −115 °C for Bio-HDPE corresponds to γ-relaxation of the green polyolefin, which is directly related to its glass transition temperature T_g_ [72], which is considered as the maximum peak value. This value is very similar to those observed in other studies [57]. A second relaxation, which is called α-relaxation, can be located between 50 and 120 °C and is associated with an interlaminar shear process. α-relaxation is often separated into two processes (α y α’) due to an inhomogeneity in crystalline regions [73]. The T_g_ change given by the addition of MAS with different compatibilizers is not so significant. In the case of Bio-HDPE/PE-g-MA/MAS and Bio-HDPE/PE-g-MA/MAS/HNTs samples, a reduction of about 3–4 °C is produced, while MLO reduces it even further (5 °C). It is logic that the 7.5 phr MLO sample possesses the lowest T_g_ due to the plasticizing effect previously mentioned. Regarding the combination of HNTs (3.75 phr) and MLO (3.75 phr), they seem to exert a synergetic effect that moves the glass transition peak +1 °C approximately. This can be associated with a good dispersion of HNTs into the polymer matrix, which immobilize Bio-HDPE chains to some extent. Only with the superposition of all the effects commented before, a significant difference could be observed, HNT/MLO blend reduced the T_g_ to the lowest temperature (−114.3 °C).

### 3.5. Color Measurement of BIO-HDPE/PE-g-MA/MAS Blends

Color, luminance, and transparency are essential issues to be considered in materials that try to imitate wood or WPC, in this case reddish-like, as those parameters determine how similar in appearance are these materials to wood. Table 6 gathers color coordinates values of Bio-HDPE/PE-g-MA/MAS blends, while Figure 8 shows the visual appearance of tensile test samples. All samples are opaque, mainly due to the semicrystalline nature of Bio-HDPE, which does not allow light to pass through [74].

L*a*b* color coordinates were measured on injection-molded samples in order to analyze the color variations and luminance. Luminance (L*) is indicative of lightness. Neat Bio-HDPE showed high brightness as a consequence of its characteristic white color, in contrast with the brown-like color of the other samples, which present low luminance as a result of the addition of fillers and compatibilizers. Referring to color coordinate a*, it is indicative of the color change between green (negative) and red (positive). Particularly, neat Bio-HDPE presents a negative value but close to 0, −2.29 in this case, because of the proximity of the sample to white color. Similar value was reported by Rojas-Lema et al. [75] in neat Bio-HDPE films. The rest of materials have positive and quite similar values each due to the characteristic brown color of micronized argan shell [76]. However, a slight change in brown tonality can be observed between those samples with and without MLO. MLO samples display a darker brown color, as a result they present a lower a* value than samples lacking MLO, which have a clearer brown color as well as closer to pure red. This can be due to the trend of MLO to take a* coordinate to more negative values, as it was observed by Quiles-Carrillo et. al. [77]. Regarding b* coordinate color, it defines blue (negative) and yellow (positive) colors. Neat Bio-HDPE has a negative value of −5.35, which is similar to that reported by Rojas-Lema et. al. [75]. While the other materials present similar values between 4 and 5, due to approaching the intrinsic yellow-like color of the argan particles and MLO. It should be noted that yellow tonality of MLO is less intense than that of argan particles, therefore reaching lower b* values. On the other hand, HNTs do not affect color significantly but seem to reduce color coordinate a* and slightly increase coordinate b*. The statistical results show that the biggest color difference was assessed by the introduction of the filler, the compatibilization strategies followed slightly provided modifications between them.

Colors showed by the materials studied here make them perfect candidates for wood plastic composites [78], presenting reddish-like brown colors that allow them to be used in wood-based products fabrication, where quality and aesthetic are vital and determine the success or failure of the product [79].

### 3.6. Wetting Properties and Water Absorption of Bio-HDPE/PE-g-MA/MAS Blends

One of the main disadvantages of green composites or any material based on hydrophilic fillers is their tendency to absorb water. With the objective of evaluating the behavior of Bio-HDPE/PE-g-MA/MAS blends against water, contact angle was measured at different times after applying a water drop over each sample surface. A great contact angle is indicative of a poor affinity toward water. Table 7 shows different contact angles for the blends at 0, 5, 10, 15, 20, and 30 min after the administration of the water drop. It can be seen as initially, all samples are hydrophobic, as their contact angles are far superior to 65°, which is the hydrophilic threshold according to Vogler [80]. This can be ascribed to the fact that the polymer matrix (Bio-HDPE) is a completely nonpolar compound, formed only by C-H bonds. This bond is formed by atoms with practically the same electronegativity. It is this reason that makes neat Bio-HDPE barely reduce its contact angle over time. When PE-g-MA and MAS are introduced into the polymer, the contact angle suffers a quick reduction (56.7° at 30 min). This can be attributed to the oxygen-based groups in PE-g-MA (anhydride groups) and to cellulose, hemicellulose, and lignin present in argan particles (hydroxyl and carbonyl groups). These groups give polarity to the material and can form hydrogen bonds with water (polar solvent), providing hydrophilicity over time [81]. Addition of HNTs into the blend provokes a faster decrease in contact angle than the previously mentioned samples. However, the contact angle at 30 min is very similar and slightly superior to that of Bio-HDPE/PE-g-MA/MAS. This could be attributed, on the one hand, to polar functional groups (hydroxyl) present in HNTs structure and, on the other hand, to their ability to retain different molecules within their tubular hollow structure, water being one of those molecules [82]. As expected, Bio-HDPE/PE-g-MA/MAS/MLO sample shows the lowest contact angle over time, reaching a considerably lower value than the rest of the samples (22.1°). This is ascribed to the high polarity of the MLO molecule, which has a great amount of anhydride and carbonyl groups, through which water can form hydrogen bonds as long as water and sample remain in contact [83]. Figure 9 perfectly reflects this behavior on the MLO sample, where the drop of water flattens over time. Finally, MLO and HNTs sample greatly reduces the hydrophobicity of the material due to the aforementioned reasons, verifying the effect of MLO over wettability.

From all these results it can be concluded that all fillers and compatibilizers used do not significantly vary hydrophilicity of Bio-HDPE at first, but they do increase water absorption in case of exposure for long periods of time. This could be corroborated by the statistical study where only some significant differences appeared at the initial time, but these were clearly accentuated over the time due to the different behaviors of the blends.

Additionally, water absorption capabilities under long time exposure were studied for each material by means of water absorption test. Figure 10 shows the water absorption of each sample after 14 weeks of immersion in distilled water. It can be observed as neat Bio-HDPE barely absorbed water, presenting an asymptotic value of 0.05 wt.-%. This remarks the hard hydrophobic nature of Bio-HDPE, as it has been previously said during the contact angle analysis. With the addition of argan particles and PE-g-MA, water absorption increases up to 0.98 wt.-%, after a period of 14 weeks. This means an increase of 94% in relation to neat Bio-HDPE. This is closely related to hydroxyl groups in lignocellulosic compounds of argan particles, which increase the facility of the material toward capturing humidity [84]. The incorporation of HNTs (7.5 phr) in the blend causes an increment in water absorption up to 1.27 wt.-%, which can be ascribed to the hollow tubular morphology of nanotubes. Their structure helps them to trap water molecules with ease, along with the effect of polar hydroxyl groups. Lastly, MLO sample achieves the highest water absorption (1.55 wt.-%) due to the plasticizing effect that it exerts over Bio-HDPE matrix, increasing its free volume and making the diffusion of water within its structure easier, as it was observed by Quiles-Carrillo et. al. [85]. As expected, 3.75 phr HNTs and 3.75 phr MLO sample presents intermediate results between those of individual HNTs and MLO samples (7.5 phr each). It should be also noted that addition of HNTs and MLO increases the water absorption speed during the first week, denoted by a more pronounced slope in the initial section showed by all three samples, in comparison with Bio-HDPE/MAS/PE-g-MA sample.

According to these results, although hydrophilicity provided by argan particles (MAS) could prove a disadvantage in some ambits, it could certainly give applicability to these materials in some other fields. One of the applications could be flowerpot fabrication, so that when the plant is irrigated, water excess is absorbed by the pot material over time.

### 3.7. Infrared Spectroscopy

Chemical composition of the injection-molded samples was analyzed by means of Fourier transformed infrared spectroscopy (FTIR). Figure 11a gathers the individual spectrums of the compatibilizers and fillers used (PE-g-MA, MAS, HNTs y MLO) from 4000 to 600 cm^−1^. First, PE-g-MA spectrum is very similar to that of Bio-HDPE, as it is a polyethylene-based (PE) compound. Absorption bands at 2840 and 2910 cm^−1^ are associated with the stretching vibration for -CH-, -CH_2_-, or -CH_3_ [86]. The peak at 1468 cm^−1^ corresponds to the deformation vibration of -CH_2_- or -CH_3_ [86]. On the other hand, the peak at 720 cm^−1^ is due to (CH_2_)_n_ rock when n ≥ 4 [86]. In the case of MAS, two bands stand out at 1030 and 900 cm^−1^, which are related to the stretching vibration of C-O and C-OH in polysaccharide rings in cellulose [87,88]. A peak at 1114 cm^−1^ can also be appreciated due to the symmetric glycosidic stretching of C-O-C bond in polysaccharide compounds of cellulose. Moreover, little absorption bands in the 1500 and 1130 cm^−1^ region can be observed, which indicate the presence of characteristic groups of cellulose, hemicellulose, and lignin. Some examples of those groups are the peak at 1227 cm^−1^, related to the stretching vibration C=O of the acetyl group in lignins [89]; and the band at 1420 cm^−1^, which corresponds to the vibration of aromatic rings in lignin [87]. The low-intensity band at 1720 cm^−1^, is ascribed to non-cellulosic compounds (pectin, lignin, and hemicellulose) [89]. Another weak peak can be found at 2900 cm^−1^, which is related to stretching vibration of C-H bonds in CH and CH_2_ groups, present in cellulose and hemicellulose [65]. A final little band is found at 3300 cm^−1^, associated to the stretching of O-H bonds in carbohydrates (cellulose and hemicellulose) [55]. Regarding the halloysite nanotubes (HNTs) spectrum, peaks at 3695 and 3619 cm^−1^ are due to the stretching vibration of Al_2_OH- (each OH is linked to two Al atoms) [90]. Bands at 3670 and 3650 cm^−1^ are ascribed to the O-H stretching of inner-surface hydroxyl groups and the out-of-phase vibration of the inner surface hydroxyl groups, respectively [90,91,92,93]. The low intensity peak at 1649 cm^−1^ is attributed to bending vibrations due to absorbed water [94], the band at 1117 cm^−1^ is due to apical Si-O bonds, while 1022 and 682 cm^−1^ bands are associated with the perpendicular stretching of Si-O-Si bonds [93,94]. Peaks at 937 and 909 cm^−1^ are consequence of the O-H deformation of inner-surface hydroxyl groups and O-H deformation of inner hydroxyl groups, respectively [93,94]. Two last bands analyzed are located at 790 and 749 cm^−1^, and can be ascribed to the O-H translation vibrations of halloysite O-H units [93]. With regard to MLO spectrum, a first peak is located at 3006 cm^−1^, corresponding to =C-H stretching of double carbon-carbon bonds, and those at 2925 and 2850 cm^−1^ refer to the antisymmetric and symmetric stretching vibration C-H of saturated carbon-carbon bonds (C-C), respectively [95]. Other bonds can be found at 1740 and 1709 cm^−1^, indicative of the C=O carbonyl stretching vibration of the ester and maleic anhydride, respectively; another band at 1158 cm^−1^ is due to the stretching vibration C-O-C, C-O, and C-C in ester groups; the peak at 720 cm^−1^ is normally related to the out-of-plane C-H stretching vibration of saturated C-C bonds [95]. Finally, the band at 1462 cm^−1^ is due to C-H bending vibration, while the absorption peaks at 1862 and 1784 cm^−1^ are attributed to anhydride groups.

Figure 11b gathers infrared spectrums for neat Bio-HDPE as well as all the developed blends. The main bands of neat Bio-HDPE are located at 2914, 2846, 1466, and 718 cm^−1^, and they are related to the stretching vibration, bending deformations, and rocking of methylene groups (CH_2_) [96]. These peaks are found in all samples, but with higher intensity, due to the presence of PE-g-MA, which is a PE-based material and as such presents the same nature as Bio-HDPE. Thus, their spectrums are very similar between each other. The weak absorption bands between 1370 and 1350 cm^−1^ are associated with wagging and symmetric deformations of CH_2_ and CH_3_ groups, respectively. The addition of MAS compatibilized with PE-g-MA makes a band in the range 1025–1091 cm^−1^ to appear, which is mainly attributed to the formation of PE-g-MA dimers or oligomers, identified with the stretching of C-C and C-O bonds [59]. Moreover, this band could be combined with another one at 1030 cm^−1^, relative to the presence of C-O and C-OH bonds in MAS particles cellulose. The presence of HNTs in the blend deforms and increases the intensity of the peak at 1030 cm^−1^, due to the perpendicular Si-O-Si stretching, as it has been previously mentioned in the individual spectra of HNTs. Additionally, in HNTs samples a weaker band at 900 cm^−1^ can be observed, ascribed to the deformation of O-H groups [90]. Moreover, little peaks can be located in the 3650 cm^−1^ region, which have been previously identified in the individual analysis of HNTs, associated with the stretching vibration of inner-surface O-H groups in the nanotubes. Regarding the MLO presence in the blend, it produces changes found at 1690 cm^−1^ approximately, related to the C=O stretching in hydrolyzed anhydride groups in MLO [13]. This peak has already been observed in MLO individual spectrum, as it has been aforementioned, and it does not appear in those samples without MLO on their structure. The appearance of low-intensity bands at 3700 and 3600 cm^−1^ can be ascribed to the absorption of water, catalyzed by MLO and argan particles, which are polar compounds with great amount of functional oxygen-based groups and great affinity toward water. HNTs and MLO blend shows the same characteristic peaks described for each individual sample, but with less absorption intensity, as their concentration reduces from 7.5 phr to 3.75 phr.

These results seem to reveal certain compatibilization between elements. This fact is denoted by a general increase in the intensity of peaks related to oxygen-based groups, indicating a possible bonding between anhydride groups in PE-g-MA and MLO with hydroxyl groups in MAS and HNTs, increasing the affinity reciprocally within the polymer matrix.

## 4. Discussion

The present work shows the incorporation of argan shell wastes in wood plastic composites with biobased polyethylene matrix and different compatibilizers. The incorporation of this type of loads can be effectively used as new reinforcement elements in order to create parts prepared by conventional industrial processes for thermoplastic materials, in special injection molding. In relation to the mechanical properties, the incorporation of MAS (30 wt.%) with PE-g-MA (3 wt.%) into the Bio-HDPE matrix increases the Young modulus up to 846 MPa. This can be directly related to a proper distribution of MAS particles in the polymer matrix, leading to a good adhesion of the filler with the polymer and obtaining a more rigid material. Addition of HNTs in the blend showed an additional enhancement of stiffness. In this context, Bio-HDPE/PE-g-MA/MAS/HNTs sample presented the highest Young modulus of all blends (1126 MPa). The incorporation of MLO in the green composites increased 100% of the elongation at break of the Bio-HDPE/PE-g-MA/MAS blend. From a morphological point of view, the incorporation of argan particles in the BioHDPE matrix, showed great cohesion, which implies good interaction between both elements. Furthermore, the addition of MLO verified the results obtained in the mechanical properties, showing a more ductile fracture and confirming the plasticizing effect of MLO. However, the incorporation of the lignocellulosic filler leads to a reduction in thermal properties and a decrease in the crystallinity of the compound. In relation to the thermomechanical properties, the incorporation of HNTs highly augmented the storage modulus, providing values of 3111 and 1898 MPa. This is indicative of an increase in the stiffness of the material, confirming the results commented in the mechanical properties section, associated with a high dispersion of HNTs in the matrix. In general, the incorporation of argan particles and additives provided the materials with reddish-brown colors that allow their use in the manufacturing of wood-based products. In relation to one of the main drawbacks of green composites, it can be seen how the incorporation of lignocellulosic particles greatly increases the water absorption of the composites, generating certain disadvantages. In addition, the incorporation of the MLO increases the water absorption capacity of the composites, due to the plasticization effect it exerts over the Bio-HDPE matrix.

## 5. Conclusions

The results obtained in this work indicate that it is possible to obtain WPC with high renewable content with Bio-HDPE, lignocellulosic fillers and natural additives. This type of green composites can greatly favor the generation of circular economies focused on giving added value to agri-food waste from the Mediterranean basin, also favoring the creation of highly efficient polymers at very competitive costs. The materials obtained in this work have proved to possess excellent properties at a reduced cost in comparison with the neat Bio-HDPE. High ductile properties, relatively high stiffness and good thermal stability were reported, as well as a visual appearance very similar to that of reddish-color woods, which is essential in a wood plastic composite. The samples showed certain hydrophilicity over time, which can prove to be a disadvantage, although it is unusual for these materials to be immersed in water for long periods of time. However, if it were the case, they have some applications too, as it is the fabrication of plant pots. All in all, the affinity between argan particles and Bio-HDPE has been successfully increased in this study with the use of compatibilizers, and their properties vastly displayed and demonstrated. This work opens a new line of research regarding the development of new materials formed by polar fillers and non-polar polymeric matrices.

## Figures and Tables

**Figure 1 polymers-13-00922-f001:**
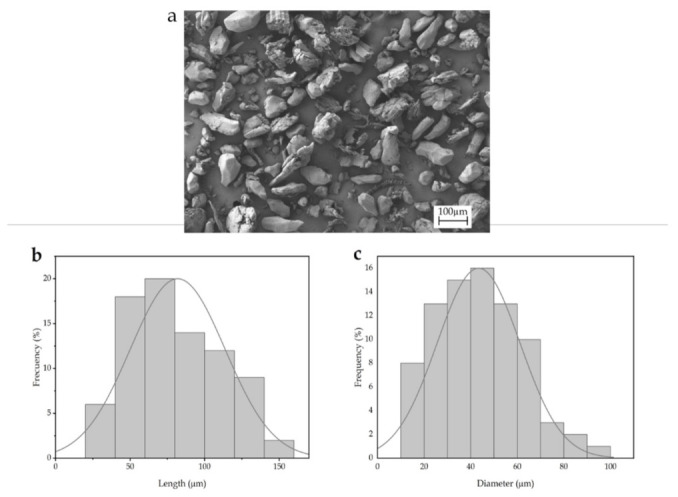
(**a**) Scanning electron microscope image (SEM) of micronized argan shell (MAS). Image was taken with a magnification of 100× and a scale marker of 100 µm; (**b**) histogram of the argan particles length; (**c**) histogram of the argan particles diameter.

**Figure 2 polymers-13-00922-f002:**
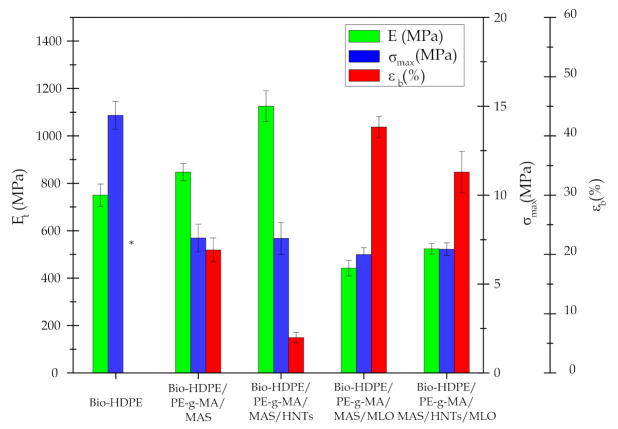
Mechanical properties of the injection-molded samples of Bio-HDPE blends. Tensile modulus (E), maximum tensile strength (σ_max_) elongation at break (ε_b_). * Elongation at break for the Bio-HDPE could not be assessed because breakage is not achieved.

**Figure 3 polymers-13-00922-f003:**
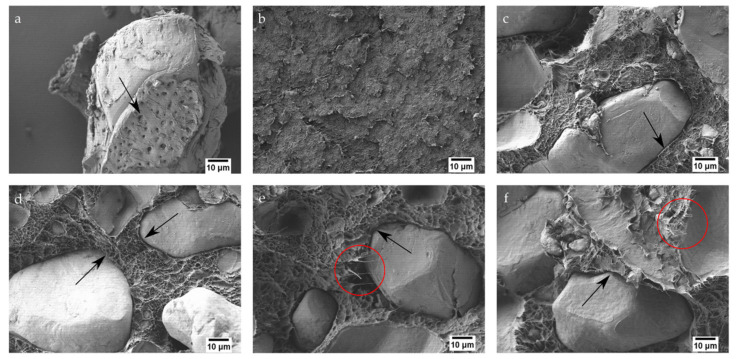
Field emission scanning electron microscopy (FESEM) images at 1000× of the fractured surfaces of: (**a**) MAS; (**b**) neat Bio-HDPE; (**c**) Bio-HDPE/PE-g-MA/MAS; (**d**) Bio-HDPE/PE-g-MA/MAS/HNTs; (**e**) Bio-HDPE/PE-g-MA/MAS/MLO; (**f**) Bio-HDPE/PE-g-MA/MAS/HNTs/MLO.

**Figure 4 polymers-13-00922-f004:**
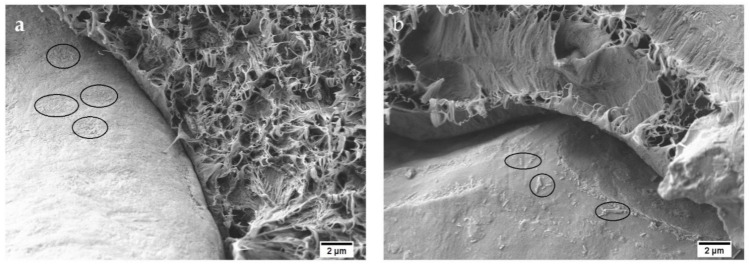
Field emission scanning electron microscopy (FESEM) images at 5000× of the fractured surfaces of samples compatibilized with HNTs: (**a**) Bio-HDPE/PE-g-MA/MAS/HNTs (7.5 phr HNTs); (**b**) Bio-HDPE/PE-g-MA/MAS/HNTs/MLO (3.75 phr HNTs).

**Figure 5 polymers-13-00922-f005:**
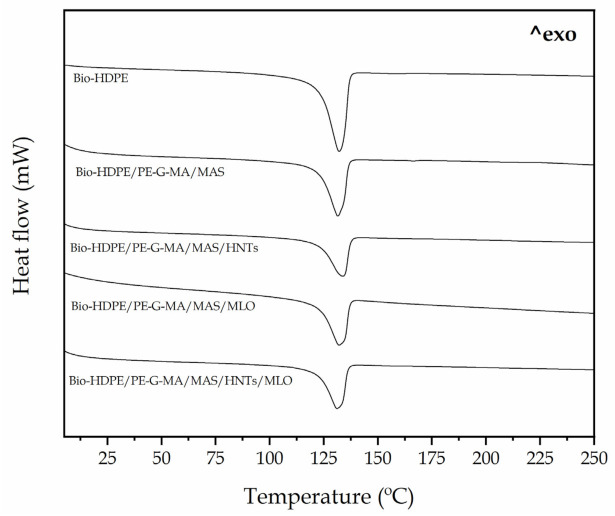
Differential scanning calorimetry (DSC) thermograms of Bio-HDPE/PE-g-MA/MAS blends with different compatibilizers.

**Figure 6 polymers-13-00922-f006:**
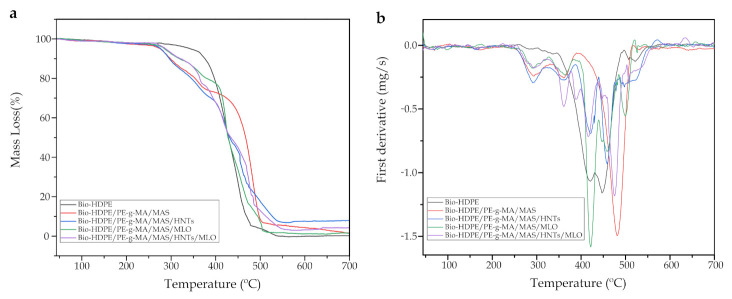
(**a**) Thermogravimetric analysis (TGA) curves and (**b**) first derivative (DTG) of Bio-HDPE/PE-g-MA/MAS blends with different compatibilizers.

**Figure 7 polymers-13-00922-f007:**
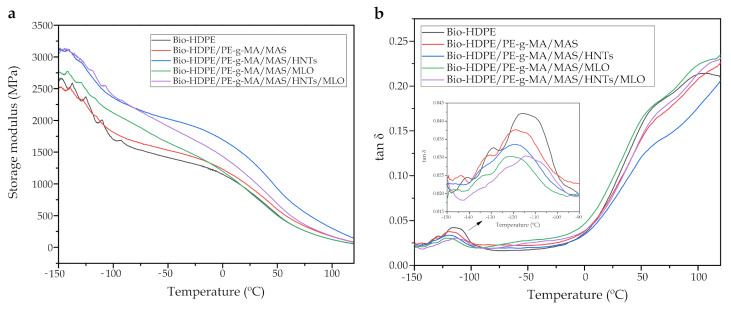
Plot evolution of (**a**) the storage modulus (*G′*) and (**b**) the dynamic damping factor (tan δ) of the injection-molded samples of Bio-HDPE/PE-g-MA/MAS blends with different compatibilizers.

**Figure 8 polymers-13-00922-f008:**
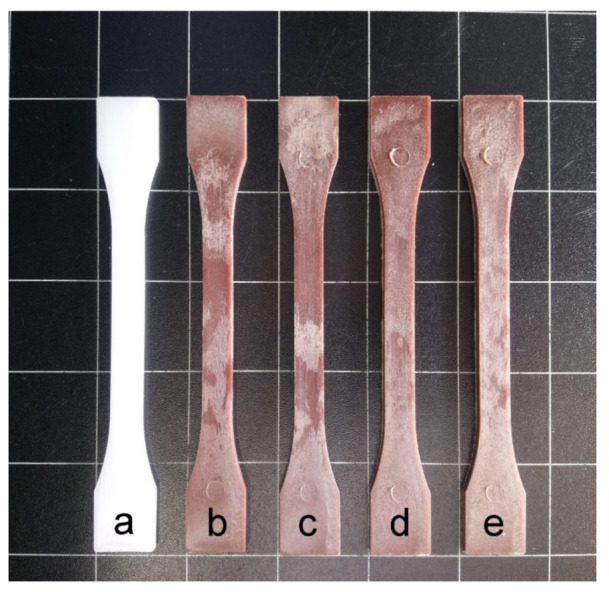
Visual appearance of the samples: (**a**) Bio-HDPE; (**b**) Bio-HDPE/PE-g-MA/MAS; (**c**) Bio-HDPE/PE-g-MA/MAS/HNT; (**d**) Bio-HDPE/PE-g-MA/MAS/MLO; (**e**) Bio-HDPE/PE-g-MA/MAS/HNT/MLO.

**Figure 9 polymers-13-00922-f009:**
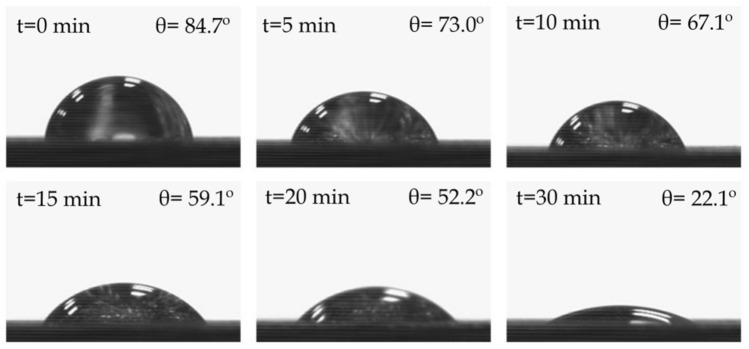
Water contact angle of the sample Bio-HDPE/PE-g-MA/MAS/MLO over time: 0, 5, 10, 15, 20, and 30 min.

**Figure 10 polymers-13-00922-f010:**
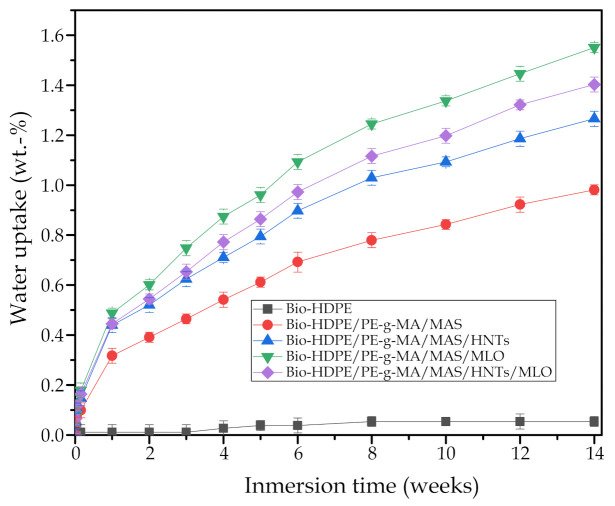
Water uptake of the injection-molded pieces of neat Bio-HDPE and its green composites with argan shell micronized (MAS) compatibilized with PE-g-MA, MLO, and with HNTs as additional filler.

**Figure 11 polymers-13-00922-f011:**
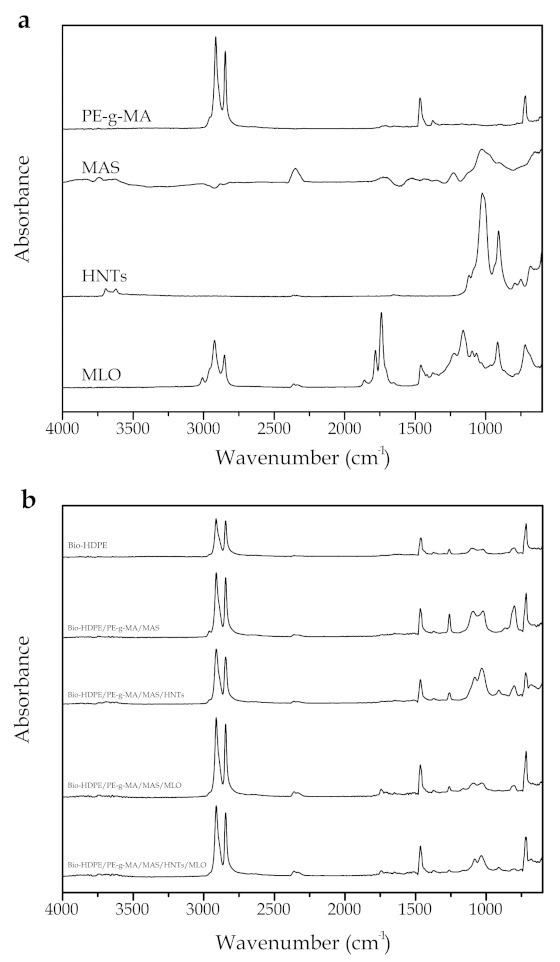
(**a**) Fourier transform infrared (FTIR) spectra, from bottom to top, of maleinized linseed oil (MLO), halloysite nanotubes (HNTs), micronized argan shell (MAS), Polyethylene-graft-maleic anhydride (PE-g-MA) (**b**) FTIR spectra of the different blends, from top to bottom Bio-HDPE, Bio-HDPE(67 wt.%)/PE-g-MA(3 wt.%)/MAS(30 wt.%),Bio-HDPE(67 wt.%)/PE-g-MA(3 wt.%)/MAS(30 wt.%)/HNTs(7.5 phr), Bio-HDPE(67 wt.%)/PE-g-MA(3 wt.%)/MAS(30 wt.%)/MLO(7.5 phr), Bio-HDPE(67 wt.%)/PE-g-MA(3 wt.%)/MAS(30 wt.%)/HNTs(3.75 phr)/MLO(3.75 phr).

**Table 1 polymers-13-00922-t001:** Summary of compositions according to the weight content (wt.%) of Bio-HDPE/MAS and different compatibilizers and additives.

Code	BIO-HDPE (wt.%)	PE-g-MA (wt.%)	MAS (wt.%)	HNTs (phr)	MLO (phr)
Bio-HDPE	100	0	0	0	0
Bio-HDPE /PE-g-MA/MAS	67	3	30	0	0
Bio-HDPE /PE-g-MA/MAS/HNT	67	3	30	7.5	0
Bio-HDPE /PE-g-MA/MAS/MLO	67	3	30	0	7.5
Bio-HDPE /PE-g-MA/MAS/HNT/MLO	67	3	30	3.75	3.75

**Table 2 polymers-13-00922-t002:** Summary of mechanical properties of the injection-molded samples of Bio-HDPE blends. Tensile modulus (E), maximum tensile strength (σ_max_) elongation at break (ε_b_), Shore D hardness, and impact (Charpy) strength.

Code	E (MPa)	σ_max_ (MPa)	ε_b_ (%)	Shore D Hardness	Impact Strength (kJ/m^2^)
Bio-HDPE	750 ± 47 ^a^	14.48 ± 0.78 ^a^	Nb ^a^	56.2 ± 1.3 ^a^	2.7 ± 0.2 ^a^
Bio-HDPE/PE-g-MA/MAS	846 ± 36 ^b^	7.57 ± 0.81 ^b^	20.7 ± 2.0 ^b^	59.2 ± 0.8 ^a^	1.4 ± 0.1 ^b^
Bio-HDPE/PE-g-MA/MAS/HNTs	1126 ± 65 ^c^	7.57 ± 0.90 ^b^	6.0 ± 0.9 ^c^	60.6 ± 1.3 ^a^	1.7 ± 0.1 ^c^
Bio-HDPE/PE-g-MA/MAS/MLO	442 ± 33 ^d^	6.66 ± 0.39 ^c^	41.5 ± 1.7 ^d^	53.2 ± 0.8 ^b^	2.2 ± 0.3 ^d^
Bio-HDPE/PE-g-MA/MAS/HNTs/MLO	523 ± 26 ^e^	6.98 ± 0.59 ^c^	33.9 ± 3.5 ^e^	54.6 ± 0.5 ^b^	2.1 ± 0.3 ^d^

^a–e^ Different letters in the same column indicate a significant difference among the samples (*p* < 0.05).

**Table 3 polymers-13-00922-t003:** Melting temperature (T_m_), melting enthalpy (∆H_m_) and crystallinity (X_C_) of Bio-HDPE/PE-g-MA/MAS blends with different compatibilizers, obtained by differential scanning calorimetry (DSC).

Code	T_m_ (°C)	∆H_m_ (J·g^−1^)	X_C_ (%)
Bio-HDPE	131.0 ± 0.5 ^a^	194.2 ± 1.5 ^a^	66.3 ± 0.9 ^a^
Bio-HDPE/PE-g-MA/MAS	130.4 ± 0.3 ^a^	145.7 ± 1.2 ^b^	49.7 ± 0.7 ^b^
Bio-HDPE/PE-g-MA/MAS/HNTs	133.1 ± 0.4 ^a^	110.3 ± 1.0 ^c^	37.6 ± 0.6 ^c^
Bio-HDPE/PE-g-MA/MAS/MLO	131.2 ± 0.3 ^a^	116.7 ± 1.2 ^d^	39.8 ± 0.9 ^c^
Bio-HDPE/PE-g-MA/MAS/HNT/MLO	130.4 ± 0.2 ^a^	118.8 ± 2.0 ^d^	40.5 ± 0.7 ^c^

^a–d^ Different letters in the same column indicate a significant difference among the samples (*p* < 0.05).

**Table 4 polymers-13-00922-t004:** Main thermal degradation parameters of the Bio-HDPE/MAS blends with different compatibilizers in terms of the onset degradation temperature at a mass loss of 5 wt.% (*T_5%_*), maximum degradation rate (peak) temperature (*T_deg_*), and residual mass at 700 °C.

Code	T_5%_ (°C)	T_deg_ (°C)	Residual Weight (%)
Bio-HDPE	342.8 ± 0.9 ^a^	447.3 ± 2.2 ^a^	0.3 ± 0.1 ^a^
Bio-HDPE/PE-g-MA/MAS	275.8 ± 1.2 ^b^	481.3 ± 1.7 ^b^	1.5 ± 0.1 ^b^
Bio-HDPE/PE-g-MA/MAS/HNT	276.8 ± 1.3 ^b^	458.3 ± 1.5 ^c^	7.8 ± 0.2 ^c^
Bio-HDPE/PE-g-MA/MAS/MLO	285.3 ± 0.8 ^b^	421.8 ± 1.8 ^d^	1.5 ± 0.2 ^d^
Bio-HDPE/PE-g-MA/MAS/HNT/MLO	287.8 ± 1.1 ^b^	475.3 ± 0.9 ^e^	4.3 ± 0.1 ^e^

^a–e^ Different letters in the same column indicate a significant difference among the samples (*p* < 0.05).

**Table 5 polymers-13-00922-t005:** Dynamic-mechanical properties of injection-molded samples of Bio-HDPE/PE-g-MA/MAS blends with different compatibilizers, at different temperatures.

Parts	E’ (MPa) at −140 °C	E’ (MPa) at −25 °C	E’ (MPa) at 100 °C	T_g BIO-HDPE_ (°C) *
Bio-HDPE	2513 ± 30 ^a^	1309 ± 14 ^a^	124 ± 2 ^a^	−115.0 ± 1.1 ^a^
Bio-HDPE/PE-g-MA/MAS	2523 ± 25 ^a^	1413 ± 16 ^b^	170 ± 4 ^b^	−118.7 ± 1.3 ^a^
Bio-HDPE/PE-g-MA/MAS/HNTs	3111 ± 39 ^b^	1898 ± 20 ^c^	299 ± 8 ^c^	−119.0 ± 2.0 ^a^
Bio-HDPE/PE-g-MA/MAS/MLO	2730 ± 31 ^c^	1449 ± 17 ^d^	127 ± 2 ^d^	−120.5 ± 3.2 ^a^
Bio-HDPE/PE-g-MA/MAS/HNT/MLO	3090 ± 37 ^d^	1685 ± 23 ^e^	176 ± 5 ^e^	−114.3 ± 1.2 ^b^

***** The T_g_ has been measured using the tan δ peak maximum criterion. ^a–e^ Different letters in the same column indicate a significant difference among the samples (*p* < 0.05).

**Table 6 polymers-13-00922-t006:** Luminance and color coordinates (L*a*b*) of the Bio-HDPE/PE-g-MA/MAS blends with different compatibilizers.

Code	L *	a *	b *
Bio-HDPE	72.7 ± 0.3 ^a^	−2.29 ± 0.01 ^a^	−5.35 ± 0.07 ^a^
Bio-HDPE/PE-g-MA/MAS	37.6 ± 0.7 ^b^	6.21 ± 0.48 ^b^	4.95 ± 0.61 ^b^
Bio-HDPE/PE-g-MA/MAS/HNT	39.2 ± 2.4 ^b^	5.76 ± 0.37 ^c^	5.05 ± 1.12 ^b^
Bio-HDPE/PE-g-MA/MAS/MLO	36.1 ± 0.2 ^b^	5.53 ± 0.31 ^c^	4.23 ± 0.06 ^c^
Bio-HDPE/PE-g-MA/MAS/HNT/MLO	37.0 ± 0.1 ^b^	5.21 ± 0.16 ^c^	4.39 ± 0.24 ^c^

^a–c^ Different letters in the same column indicate a significant difference among the samples (*p* < 0.05).

**Table 7 polymers-13-00922-t007:** Contact angle (*θ**_w_*) of different Bio-HDPE/PE-g-MA/MAS blends with different compatibilizers at several times of exposure to water: 0, 5, 10, 15, 20 and 30 min.

Code/Time	0 min	5 min	10 min	15 min	20 min	30 min
Bio-HDPE	90.1 ± 3.2° ^a^	88.1 ± 2.5° ^a^	88.2 ± 1.3° ^a^	84.5 ± 4.2° ^a^	83.3 ± 1.0° ^a^	81.1 ± 2.1° ^a^
Bio-HDPE/PE-g-MA/MAS	89.9 ± 2.4° ^a^	80.0 ± 0.8° ^b^	78.6 ± 2.0° ^b^	76.1 ± 1.7° ^b^	72.2 ± 0.6° ^b^	56.7 ± 0.5° ^b^
Bio-HDPE/PE-g-MA/MAS/HNTs	87.6 ± 1.3° ^a^	77.9 ± 0.9° ^b^	70.1 ± 1.8° ^c^	67.7 ± 1.3° ^c^	61.0 ± 0.9° ^c^	59.4 ± 0.7° ^c^
Bio-HDPE/PE-g-MA/MAS/MLO	84.7 ± 0.9° ^b^	73.0 ± 1.1° ^c^	67.1 ± 1.7° ^c^	59.1 ± 0.8° ^d^	52.2 ± 0.6° ^d^	22.1 ± 0.4° ^d^
Bio-HDPE/PE-g-MA/MAS/HNTs/MLO	92.5 ± 3.1° ^c^	81.4 ± 1.9° ^d^	78.8 ± 1.2° ^d^	73.8 ± 1.3° ^e^	66.4 ± 0.8° ^e^	47.7 ± 0.8° ^e^

^a–e^ Different letters in the same column indicate a significant difference among the samples (*p* < 0.05).

## Data Availability

Not applicable.

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
