# Peer review of "Upgrading Argan Shell Wastes in Wood Plastic Composites with Biobased Polyethylene Matrix and Different Compatibilizers"

_polymers, 2021, doi:10.3390/polym13060922_

Round 1

Reviewer 1 Report

See the attached document, which is the review for the Authors.

Author Response

In order to simplify the response to reviewer, the suggestions have been grouped into a global response, since most of them can be answered together.

The methods used and the experimental one have followed the premises of the articles published in the field of polymeric materials, and in particular the journal Polymers. However, more description has been added to facilitate its understanding. In relation to the introduction, abstract, conclusions and background information, the authors have been added more information about Wood plastic composites (WPC) and circular economy in order to define more clearly the objective of the study.

As for the suggested papers and their structure, these are not from the field of polymers and cannot be framed in this special issue, which is Biopolymers from natural resources. For this reason, the structure of the article follows the latest articles published in this special issue. An additional Figure has been added to help the understanding of the results.

In the polymers template for the paper, the discussion must be done but conclusions are not mandatory. Nevertheless, following the recommendations some conclusions have been added for a better understanding.

Finally, possible mistakes have been revised, checked and corrected. These have been highlighted with yellow colour. In addition, an in depth check of the grammar and spelling has been carried out and all detected mistakes have been corrected. Changes done to manuscript have been emphasized in yellow in order to facilitate their searching. We have worked in accordance with all reviewer comments so we consider that the version we are sending to you includes all necessary changes.

Reviewer 2 Report

please see my comments in the attached file 

Author Response

We are sending the revised version of the manuscript that includes all suggestions and corrections proposed by the reviewer. In addition, an in depth check of the grammar and spelling has been carried out and all detected mistakes have been corrected. Changes done to manuscript have been emphasized in yellow in order to facilitate their searching. We have worked in accordance with all reviewer comments so we consider that the version we are sending to you includes all necessary changes.

  • please add (a)

Thank you for the recommendation, in the previous version (a) was included in the image. The main problem was the colour employed and the position, it was white and it was placed inside the image making difficult to see it. In the revised manuscript (a) has been moved an now its in colour black so the readers can see it more easily.

  • It would of great help for the readers and also to you in order to evaluate your results better, if you perform a statistical analysis on the data depicted in table 2. With one-way analysis (ANOVA) you will be able to identify significant differences between the treatments. Your discussion is very good and this will improve it to a greater extent

This is a great recommendation to improve the quality of the work. The ANOVA analysis has been performed in all the numerical results provided in this work. Also extra comments related with the ANOVA results have been included and highlighted in yellow.

  • An excellent session. Please show us your observations with arrows inside the figure 2 and 3.

To make more easy to understand the phenomena mentioned in the texts, arrows and circles have been included in the Figures so the readers can understand more easily the information.

  • I can not understand why you did not add conclusions. Please do so !!!

In the polymers template for the paper, the discussion must be done but conclusions are not mandatory. Nevertheless, following the recommendations some conclusions have been added for a better understanding.

Reviewer 3 Report

In the page 02. , to eliminate the word ( acronym) (PS), after polyester

Author Response

Following the recommendations by the reviewer the acronym PS has been erased in the text to avoid possible confusions. Furthermore, possible mistakes have been revised, checked and corrected. In addition, an in depth check of the grammar and spelling has been carried out and all detected mistakes have been corrected. Changes done to manuscript have been emphasized in yellow in order to facilitate their searching. We have worked in accordance with all reviewer comments so we consider that the version we are sending to you includes all necessary changes.
